# Protein Is an Intelligent Micelle

**DOI:** 10.3390/e25060850

**Published:** 2023-05-26

**Authors:** Irena Roterman, Leszek Konieczny

**Affiliations:** 1Department of Bioinformatics and Telemedicine, Jagiellonian University—Medical College, Medyczna 7, 30-688 Kraków, Poland; 2Chair of Medical Biochemistry, Jagiellonian University—Medical College, Kopernika 7, 31-034 Kraków, Poland

**Keywords:** information, bit, genetic code, translation, protein folding, information deficiency, reaching the goal, homeostasis, complexity, DNA, proteome construction

## Abstract

Interpreting biological phenomena at the molecular and cellular levels reveals the ways in which information that is specific to living organisms is processed: from the genetic record contained in a strand of DNA, to the translation process, and then to the construction of proteins that carry the flow and processing of information as well as reveal evolutionary mechanisms. The processing of a surprisingly small amount of information, i.e., in the range of 1 GB, contains the record of human DNA that is used in the construction of the highly complex system that is the human body. This shows that what is important is not the quantity of information but rather its skillful use—in other words, this facilitates proper processing. This paper describes the quantitative relations that characterize information during the successive steps of the “biological dogma”, illustrating a transition from the recording of information in a DNA strand to the production of proteins exhibiting a defined specificity. It is this that is encoded in the form of information and that determines the unique activity, i.e., the measure of a protein’s “intelligence”. In a situation of information deficit at the transformation stage of a primary protein structure to a tertiary or quaternary structure, a particular role is served by the environment as a supplier of complementary information, thus leading to the achievement of a structure that guarantees the fulfillment of a specified function. Its quantitative evaluation is possible via using a “fuzzy oil drop” (FOD), particularly with respect to its modified version. This can be achieved when taking into account the participation of an environment other than water in the construction of a specific 3D structure (FOD-M). The next step of information processing on the higher organizational level is the construction of the proteome, where the interrelationship between different functional tasks and organism requirements can be generally characterized by homeostasis. An open system that maintains the stability of all components can be achieved exclusively in a condition of automatic control that is realized by negative feedback loops. This suggests a hypothesis of proteome construction that is based on the system of negative feedback loops. The purpose of this paper is the analysis of information flow in organisms with a particular emphasis on the role of proteins in this process. This paper also presents a model introducing the component of changed conditions and its influence on the protein folding process—since the specificity of proteins is coded in their structure.

## 1. Introduction

Traditionally, the concept of information when applied to the field of biology is associated with the DNA strand and the genetic information recorded in it. In the clear majority of biochemistry studies, the focus is on the energy side of biological processes. It turns out, however, that all of these processes can be analyzed based on the interpretation of information, including—in particular—the content and flow of information as well as information processing.

The basic definition enabling evaluation of the amount of information carried by an event with a probability of p_i_ is found in the definition proposed by Shannon [1]:I = −log_2_(p_i_).(1)

This defines a unit of information as one bit for an event with p_i_ = ½. 

Based on this definition, it is possible to show the relationship between the individual stages of the “biological dogma” from the perspective of the level of information recorded.

With the emergence of the new discipline of Systems Biology, which aims to simulate the functioning of a living organism, a definition is needed for its general rules, including rules that govern regulating the processing of information [2,3,4,5,6,7,8].

This paper discusses these types of biological processes from the perspective of information flow and processing and by treating DNA as the main source of information. The critical step of high information deficiency, which is the structurization of proteins, appears supported by the additional source of information coming from the environment, such as from—in particular—water. Life without water is impossible. We present a proposed model introducing the component of changed conditions and its influence on the protein folding process. This encoded form of information in the 3D structure also makes the protein an “intelligent” micelle with the ability to perform highly specific tasks. In this study, a proposed model of a mechanism that produces tools and machines (i.e., complex protein structures) is presented, and this proposed model will also account for recreating the highest degree of organization in the structure of the entire organism—the proteome construction.

## 2. Stages Contained in the Biological Dogma

The traditional biological dogma denotation is as follows:DNA → mRNA → PROTEIN.

However, this can be extended to the following form:$$DNA\frac{\to }{1}mRNA\frac{\to }{2}AA\frac{\to }{3}3DSTRUCTURE\frac{\to }{4}FUNCTION$$

These stages, apart from energy analysis (as it is presented in most biochemistry handbooks), can be considered in the principle of operating on, and the processing of, information.

1.Step 1—Amount of Information in DNA

The amount of information in human DNA is 3 × 10^6^ × 2 [bit] = approximately 1 GB (assuming p_i_ = ¼ for every nucleotide). The assessment of this amount of information—the INPUT—may be conducted in relation to the content of the product—the OUTPUT—of which both occur in a functioning human body (We hypothesize that it is also the most complex system operating on our planet). The disproportion between the amount of information in DNA and the inconceivably high complexity of the final product that is the human body is evident.

The DNA → mRNA stage involves the selection of sections that carry information (genes), i.e., the identification and utilization of only the sections that carry the information, which are also the sections where p_i_ is different from ¼ (assuming random appearance of a particular nucleotide).

2.Step 2—mRNA → AA

Based on the definition of Equation (1), the amount of information carried by an amino acid is 4.32 bits (assuming the probability of occurrence of every amino acid = 1/20). Compared with the six bits carried by a nucleotide triplet, the degeneracy of genetic code becomes obvious. This stage is deterministic in nature. The amount of information carried by a particular amino acid is based on the frequency of occurrence of a given amino acid in the proteins that are available in the Protein Data Bank (PDB); We extracted a non-redundant subset (Figure 1—blue line) [9,10].

### 2.1. Step 3: Amino Acid Sequence (AA) → 3D Structure

This stage is critical to the production of the appropriate structure that fulfills an associated biological function.

#### 2.1.1. Interpretation of Phi, Psi Angles Distribution on Ramachandran Map

Providing the appropriate set of Phi and Psi angles requires the selection of one point on the Ramachandran map. The probability of indicating the correct set of Phi and Psi angles is 1/(359 × 359). For accuracy, 1 deg x 1 deg requires nearly 17 bits (as calculated according to Equation (1)). At the 4.32 bits level, which is carried by an amino acid, the requirement for specifying the appropriate conformation-determining angles points to stage AA → 3D as the stage with a significant information deficit. The limitation arising from the variation of the occurrence frequency of appropriate conformations (i.e., the areas preferable to energetically excluded areas). At an accuracy level of 5 deg(Phi) × 5 deg (Psi), it reduces that requirement to a level of 5–7 bits. However, this still indicates a significant deficit of information that is being carried by the given amino acid, even in relation to the requirement for indication/selection of a non-satisfactory determination accuracy of an appropriate conformation (Figure 1) [10].

#### 2.1.2. Additional Source of Information: The Environment

The conformation of individual amino acids is obtained through the action of the internal force field (i.e., the non-bonding interaction between amino acids in the chain). The information deficit indicated in the previous subsection is complemented by the active participation of the environment, which is an external force field in the production of appropriate structures that can be treated as tools/machines to perform precisely defined biological tasks [11,12,13,14].

The partner that actively participates in the folding process is water, which conditions the biological activity of every living organism. Water—as a supplier of the external field in which biological processes take place, including protein folding in particular—is not recognized except in the structure of ice [15]. Water, as an immanent component of all life processes, is rarely an object of analysis in itself. However, numerous studies have analyzed changes in the characteristics of the aquatic environment depending on the presence of external components [15,16,17,18,19,20,21,22,23,24,25,26,27,28,29,30,31,32,33,34].

The effect of water’s participation in structurization processes is the generation of micelle formed by bi-polar molecules. A highly ordered system of spherical micelles is obtained by directing the hydrophobic parts of these molecules toward the center of the structure and isolating them from the polar surroundings by a surface layer that is composed of the polar fragments of these molecules. Assuming that the amino acids constitute a set of 20 different bi-polar molecules with different proportions of the hydrophobic part in relation to the polar part, it can be assumed that the function of the set of amino acids is to achieve an enthalpy–entropic effect that is similar to that seen in the structuring of micelles. This novel conceptualization of the hydrophobic nucleus as stabilizing the tertiary structure, as presented in this paper, is an eloquent expression.

Therefore, a 3D Gaussian function was used to describe the hydrophobicity distribution, thereby expressing the concentration of hydrophobicity in the central part of the protein and the polar surface, which are composed of polar amino acids:(2)$$H_{i}^{T}=\frac{1}{H_{sum}^{T}}\exp\left(\frac{-{\left(x_{i}-\bar{x}\right)}^{2}}{2\sigma_{x}^{2}}\right)\exp\left(\frac{-{\left(y_{i}-\bar{y}\right)}^{2}}{2\sigma_{y}^{2}}\right)\exp\left(\frac{-{\left(z_{i}-\bar{z}\right)}^{2}}{2\sigma_{z}^{2}}\right).$$

The parameters σ_x_, σ_y_, and σ_z_ are adapted to the dimensions and shape of the protein. 

The 3D Gaussian function spread over the body of the protein, the values of which are assigned to the positions of the effective atoms (i.e., the averaged position of the atoms that compose the amino acid), represents the idealized hydrophobicity distribution, assuming that the protein recreates the micelle structure. On the other hand, one shall assume that this system is not necessarily reproduced in every protein. With this fact in mind, the level of hydrophobicity assigned to each effective atom, which is the result of hydrophobic inter-amino acid interactions, is determined. Here, the function proposed by Levitt [35] was applied:(3)$$H_{i}^{O}=\frac{1}{H_{sum}^{O}}\sum_{j}\left\{\begin{array}{l} \left(H_{i}^{r}+H_{j}^{r}\right)\left(1-\frac{1}{2}\left(7{\left(\frac{r_{ij}}{c}\right)}^{2}-9{\left(\frac{r_{ij}}{c}\right)}^{4}+5{\left(\frac{r_{ij}}{c}\right)}^{6}-{\left(\frac{r_{ij}}{c}\right)}^{8}\right)\right)\;for\;r_{ij}\leq c\\ 0,\;\;for\;r_{ij}>c \end{array}\right..$$

These interactions are dependent on the intrinsic hydrophobicity of (*H^r^*) interacting with the amino acids and the distance between them (*r_ij_*). Symbol “*c*” defines the cutoff distance, which is usually assumed to be 9 Å (according to [35]). As a result, each amino acid is described with two values expressing the level of hydrophobicity: *T_i_*: idealized and *O_i_*: observed. After normalization, these distributions can be compared quantitatively, thereby determining the degree of restoration regarding the T distribution by the O distribution. For this purpose, a divergence entropy [36] was applied:(4)$$D_{KL}(P|Q)=\sum_{i=1}^{N}P_{i}\log_{2}\frac{P_{i}}{Q_{i}},$$
where *P_i_* and *Q_i_* represent distribution under consideration and observed distribution, respectively. In the analysis proposed here, the role of the P distribution is fulfilled by the *O* distribution, and the role of the reference distribution *Q* is performed by the T distribution.

However, the *D_KL_* value for the *O|T* relation as the value of entropy cannot be interpreted. Therefore, a second reference—the *R* distribution—was introduced, whereby each amino acid represents the same level of hydrophobicity of *Ri* = 1/*N*, where *N* is the number of amino acids in the protein. The *R* distribution represents a state devoid of any variation in hydrophobicity levels within a protein and, thus, the opposite of a centric nucleus.

Re-determining the *D_KL_* value, this time for the *O|R* relation allows for a quantitative assessment of the “proximity” of the *O* distribution to the *R* distribution. A comparison of the *D_KL_* values for the *O|T* and *O|R* relations enables an assessment of the degree of restoration by the *O* distribution of the *T* distribution or the *R* distribution. The *D_KL_* for (*O|T*) < *D_KL_* for (*O|R*) suggests the proximity of the *O* distribution to the *T* distribution. Relative Distance (*RD*) can quantitatively express the proximity of the *O* distribution versus the *T* and *R* distributions:(5)$$RD=\frac{D_{KL}(O|T)}{D_{KL}(O|T)+D_{KL}(O|R)},$$
where *D_KL_*(*O|T*) denotes the *D_KL_* value for (*O|T*) relation and *D_KL_*(*O|R*) denotes the *D_KL_* value for (*O|R*) relation. *RD* < *0.5* indicates the presence of a centric hydrophobic nucleus, while *RD* > *0.5* signifies that no hydrophobic nucleus is present (Figure 2).

It is possible to distinguish a set of proteins that perfectly satisfies the conditions of the *O* distribution. Distribution *O* is highly similar to the *T* distribution. However, the *RD* values for these proteins are very low. These proteins are from the group of fast-folding, ultra-fast-folding, and down-hill proteins (proteins reaching the energetic minimum in one step without any energy barrier) [37]. In experimental conditions, these proteins undergo reversible and multiple unfoldings. This phenomenon of reversible unfolding can be interpreted as a process of micellization that is dependent on water’s presence in its surroundings. Polar water directs hydrophobic amino acids toward the center, thus separating them from the aquatic environment by a polar surface layer with a favorable entropy–enthalpy system. This suggests micellization during protein folding as the effect of aquatic environment participation. The abovementioned proteins precisely show the hydrophobicity distribution with a central hydrophobic nucleus and a polar surface that corresponds to the spontaneous formation of micelles that are composed of bi-polar molecules. Furthermore, these are amino acids with various polarity/hydrophobicity relations.

Many enzymes of defined specificity also emerge as a result of internal interactions and an aquatic environment [13,37]. Their sequences (as opposed to those of the aforementioned groups) exclude any possible generation of the micelle structure to a degree that fully satisfies the conditions of an ordered spherical micelle. Enzymes exhibit the correct micelle structure when the residues that do not match the distribution present in the micelles are removed from the calculation of D_KL_. The residues that are locally disrupting the micelle-like system appear to be catalytic residues and will affect their immediate environment. The remaining part of the enzyme molecule satisfies the conditions of micelles, which guarantees its solubility in an aqueous medium. Hence, we conclude that the structure and function are determined in an amino acid sequence. Moreover, the folding chain follows the micellization process; it achieves this to the extent that it is optimal for a given sequence, thereby introducing local disorder and, as a result, generating specificity.

As a consequence of this maladjustment to micelle-like structuring (hydrophobicity decomposition), proteins act as “intelligent micelles”. Apart from being soluble in an aqueous medium, the protein micelle carries information in the form of local maladjustment to a system that fully reproduces the spherical micelle with a hydrophobicity distribution that is expressed by a 3D Gaussian function that is spread on the protein body. The biological certainty that the 3D structure of a protein is encoded in its sequence can be supplemented with the following statement: In the amino acid sequence, the different degrees of possibility by which to generate the micelle-like construction are determined. The degree of discordance is the measure of specificity. An idealized micelle is mainly characterized by high solubility. The local micelle-like disorder carries information about the specificity of a given protein. The intermediate RD value reflects the amount of information determining the degree of maladjustment that, in turn, is a record of its specificity. The idealized micelle is only one, while the forms of maladjustment are many and very variable. This is why many specificities can be coded by their different forms. This hypothesis is supported by the analysis presented later in this paper.

The RD calculation is accessible upon request on the CodeOcean platform: https://codeocean.com/capsule/3084411/tree (accessed on 11 May 2023), (please contact the corresponding author to obtain access to your private program instance).

The application, which was implemented in collaboration with the Sano Center for Computational Medicine (https://sano.science (accessed on accessed on 11 May 2023)) and runs on the resources contributed by ACC Cyfronet AGH (https://www.cyfronet.pl (accessed on 11 May 2023)) in the framework of the PL-Grid Infrastructure (https://plgrid.pl (accessed on 11 May 2023)), provides a web wrapper for the abovementioned computational component and is freely available at https://hphob.sano.science (accessed on 11 May 2023).

#### 2.1.3. Strategies of Representing Information Deficiency at the Protein Folding Stage

The aquatic environment is the information provider for the folding protein, eliminating the deficit between the amount of information carried by amino acids in the polypeptide chain versus the need to generate an appropriate conformation. The water environment directing the polypeptide chain toward micellization reduces a large number of possible folding paths. The aquatic environment is not the only environment in which proteins are active. A completely different environment is provided by the cell membrane which requires the exposure of hydrophobicity on the surface and, in the case of proteins, serves as a channel in which the polarity is in the central part of the protein. To describe such conditions, a complementary function for the 3D Gaussian function is used in the following form:*Mi* = *Tmax* − *Ti*,(6)
where *T_max_* is the maximal value in the *T* distribution, and *T_i_—*theoretical hydrophobicity attributed to *i*-th amino acid. The *Mi* distribution describes a situation involving the exposure of hydrophobicity with a polar center.

However, when analyzing the structures of membrane proteins, we have discovered that their description also requires the presence of a water-based field.

Hence, the final form of the *Mi* field is as follows:*M_i_* = *T_i_* + [*K* × (*T_max_* − *T_i_*)*_n_*]*_n_*,(7)
where *T_i_* is the theoretical hydrophobicity at the *i*-th amino acid, *T_max_* is the maximal value as appears in the T distribution, and index “*n*” denotes normalization. However, the *T_max_-T_i_* is for the inverse distribution (as was assumed for the cell-membrane environment). Index “*n*” represents normalization. The *K* parameter serves a very important role in the definition of the *M* field: it denotes the degree of involvement regarding the factor modifying the specificity of the polar water field. An analysis of the multiple proteins reveals the need for modification as expressed by the component *K* × (*T_max_* − *T_i_*)*_n_*. The effect of the *K* ≠ 0 coefficient’s presence is shown in Figure 3. The selection of the correct *K* value is related to the minimum *D_KL_* value for the relation (*O|M*) (Figure 4).

### 2.2. Environment Participation in the Folding Process

The abovementioned proteins (fast-folding, ultra-fast-folding, and down-hill) exhibit a structuring described by *K* = 0. This means that information from the aquatic environment is sufficient for their structuring. To support the presented model, examples visualizing the role of the environment expressed by different *K* values are shown below.

#### 2.2.1. Protein Representing a Structure Consistent with the FOD *K* = 0 Model

One example in this group of proteins (fast-folding, ultra-fast-folding, and down-hill) is a protein from the anti-freeze protein group (PDB ID—1B7I [38]) where *RD* = 0.289 and *K* = 0.1. Negligible differences in the fit obtained for *K* = 0.1 are visible for the set of profiles *T*, *O*, and *M* that reveal a hydrophobicity arrangement that is highly consistent with the 3D Gaussian distribution. Antifreeze proteins based on the FOD model do not interact with ice-like protein–ligand (docking procedure) compounds (as is interpreted in numerous works [39,40,41]), but act on a principle comparable to the action of ions (e.g., salt applied to the ground in the winter season). Their role consists of imposing, through the polar surface of the protein, an arrangement of water molecules that differs from that found in the structure of ice. The most important functional feature of this protein is its solubility. The 3D Gaussian hydrophobicity distribution guarantees this very feature (Figure 5 shows the low *Ti* values that are consistent with low *Oi* values) by exposing the polar residues.

#### 2.2.2. A Protein Representing a Local Maladjustment to the Micelle-Like System

One example of a protein with higher values of *RD* and *K* is the lysozyme. Due to these higher values, it is representative of the lysozyme enzyme group (PDB ID 1LZ1 [42]). A scan of the *T*, *O*, and *M* profiles reveals a high degree of similarity with the exception of a few residues (Figure 6). The status of this protein is described by the parameters *RD* = 0.529 and *K* = 0.5 (Figure 6). A local hydrophobicity deficit is visible along sections 53–60. This is the section forming the substrate binding cavity. Elimination of residues deviating from the calculation (e.g., a high difference between *Ti* and *Oi* distributions) results in the values of *RD* = 0.493 and *K* = 0.4. These residues are highlighted in Figure 6. The residues disturbing the arrangement according to the 3D Gauss distribution are 35Glu and 53Asp (a local hydrophobicity deficit) and 128C (local hydrophobicity excess). The residues 35Glu and 53Asp are catalytic residues. On the other hand, 128C (a component of the disulfide bond), which is located on the surface together with catalytic residues, provides a record of information concerning the specificity of this enzyme. Inadequacy (local hydrophobicity exposure) is a source of information sent to the environment and most likely results in the appropriate ordering/disordering of water in the immediate vicinity thereby acting as a signal for the substrate.

#### 2.2.3. Periplasmic Environment

Another example is a protein acting in periplasmic space (PDB ID 2LGN [43]). The *T*, *O*, and *M* profile set visualizes the mismatch between the *T* and *O* distributions (*RD* = 0.610, *K* = 0.7), which applies to the entire chain (Figure 7).

The positions highlighted in the profiles (Figure 7A—blue) and in the 3D structure of the residues (Figure 7B) reveal elevated levels of hydrophobicity on the surface, while the residues highlighted in orange (in both the profiles and in the 3D presentation) show a local excess in the area, which is where low levels of hydrophobicity are expected. In this case, eliminating residues whose status is not adapted to a micelle-like system is not possible. The mismatch between the O distribution and the *T* distribution applies to the entire chain. The *M* distribution for *K* = 0.6 differs considerably from the *T* distribution. This means that the folding protein adapts to the environment and adopts a structure that could not be formed in an aqueous environment.

#### 2.2.4. Membrane Environment

An example in which there is a considerable dissimilarity between the *O* distribution and the *T* distribution, due to different environments, is found in transmembrane proteins [44,45]. A representative of this group is rhodopsin (PDB ID 3QAP [46]). The combination of *T*, *O*, and *M* profiles for this protein reveals significant deficits in the central part of the molecule, i.e., the retinal binding site and hydrophobicity exposure along the surface sections of the chain. The differences between the *T* and *O* distributions are expressed by the high values of *RD* = 0.777 and *K* = 1.3, respectively (Figure 8).

The juxtaposition of the proteins presented above, characterized by the increasing participation of non-aqueous compounds, highlights the role played by the environment in shaping their structure and, thus, ensuring their biological activity. However, in all the *M* distributions, the share of the distribution of the 3D Gaussian function is present; this means that the presence of water and its impact on the formation of the structure is of critical importance. Hence, a full or limited tendency toward micellization exists, albeit one that is modified to varying degrees by other environmental components.

Protein folding (based on the DNA coding system with the additional source of information from the environment) produces simple proteins of well-defined specificity. The next step of information processing is the construction of proteins, such as chaperonins, for example. However, the proper functioning of an organism becomes possible if all protein properly take place in the next step of higher system construction: the proteome (see Section 2.4).

The influence of water changed by other compounds (or by physical processes, such as shaking) has been the subject of numerous studies [15,16,17,18,19,20,21,22,23,24,25,26,27,28,29,30,31,32,33]. Shaking as a process that produces amyloid forms of various proteins (introducing a much higher share of air/water interphase), and as has been shown in experimental research, disturbs the standard order of water thereby facilitating the transformation of a protein structure into a form that favors the structure of multi-chain fibril complexes. The influence of many other compounds on structural changes (considering urea causing protein unfolding) most likely results in changes in the structure of water itself. The weakening of its impact on protein structuring is evident here (as in the example of urea), all the more so as the removal of the denaturing compound (urea dialysis) results in a return to the native structure. This is because the structure of the water is restored to its standard form.

### 2.3. Step 4: 3D Structure → FUNCTION

The term “biological function” may be substituted with the phrase “achieving a goal”. The ways in which a goal can be achieved are through the use of energy (in a probabilistic system) or through information (in a deterministic system) [15]. Proteins as tools are responsible for the vast majority of processes. In other words, they are used to achieve specific goals. How a task is accomplished depends on the predictability of the course of a given process. Generally, a particular system is used for probabilistic tasks and another for deterministic processes.

Every process in the body accomplishes a particular goal. Therefore, each process requires a mechanism that guarantees a strictly defined goal. The probability of achieving this goal is expressed by the formula:P = 1 − (1 − p)^k^,(8)
where P is the probability of achieving the goal, p is the probability of an elementary event, and k is the number of repetitions to increase the probability of P.

Achieving a certain goal (P = 1) with a very low p is possible by increasing the number of repetitions k. In a lottery draw, p, the probability of drawing the winning numbers tends to be very low. Increasing the number of tickets sold increases the probability of winning.

Another way of achieving this goal is to increase the *p*-value. In the case of the lottery game model, to increase the value of *p*, we need to know the rules of the game. The aim is to modify the lottery system in such a way that “our” selected numbers are obtained as a result of a distorted draw. For example, we magnetize the balls and manipulate how they come out of the tumbler.

Juxtaposing two solutions to help achieve a goal by using both paths is unethical, as it is an idea based on the conduct of war. Achieving the goal using high k involves firing a huge number of missiles in the direction of, for example, a flying plane (*p* has a very low value). The same goal is achieved by increasing the value of *p* via introducing a cruise missile. The difference lies in the information that the cruise missile contains regarding its destination target, which the traditional missile lacks.

The two aforementioned scenarios for achieving the same goal are fundamentally different. The ENERGY outlay in the case of path “k” is enormous while by increasing “p” we invest in INFORMATION about the system and the target at which the missile is aimed. Here, the energy input is much less. We produce a more expensive missile but only one for each object. Additionally, path k is the path along which the **addressee is unknown**, while in the case of path p, the specific addressee is **very well defined**.

Below, we present three examples representing practical adaptation of the strategy based on a large number of attempts and the information-based strategy leading to reaching the same goal. All of these examples involve biological systems. The first example involves the strategy used by plants to propagate. In the second example, the same problem in humans can be interpreted by considering the p and k. The third example concerns the molecular process of achieving resistance against antigens.

The first example, from the field of biology, involves the very simple strategy for generating plants based on the sowing (by wind) of a large number of seeds without knowing the addressee (a place that is favorable for the development of a new plant) and in expending an enormous amount of energy. Furthermore, the production of a very large number of seeds, path p also occurs in the plant world as a solution for rhizome (rootstalks) propagation. The rhizome tests its own location by carrying out all the necessary life processes and then changes its location, choosing a direction with better external conditions. This is the form that the act of addressing assumes in this process. As the second biological example, human reproduction is likewise based on the k principle (a large number of sperm where the addressee’s location is unknown). Human reproduction via the p path occurs in the in vitro technique. This process can be distilled into specifying the addressee. However, the condition for this technique is a recognition of the process itself, and it is this investment, i.e., increasing the value of p in the form of information about the process itself, that we wish to influence.

The third example concerning the molecular process using a strategy based on a large k value represented by the functioning of the immune system. The body does not know the addressee because it does not know the antigen it will be fighting. Therefore, a large number of antibody “specificities” are synthesized (a process that is similar to choosing numbers in a lottery game). The greater the k number, the higher the probability value of P. In this case, path p is a vaccine. It guarantees that the pool of antibodies is selected randomly by synthesis and that it contains one code that will recognize a dangerous infectious disease. The condition, i.e., the knowledge of the addressee, is the disease entity that poses a threat to the human body.

The dependence of achieving certainty (P = 1) for different k and p strategies is shown in Figure 9.

In biology, achieving a goal along the path that is based on increasing the value of *p* is realized whenever the word “specificity” occurs. The path p is realized by all enzymes, receptors, and even structural proteins (e.g., cytoskeleton, a specific principle of building linearly propagating microtubules, or “microfilament-type” complexes) or material storage proteins (e.g., lipid binding proteins or iron storage). The presented model suggests treating the level of specificity as equivalent to the level of the “intelligence” that is proposed in the title of this paper. The ability to achieve a specific goal can be treated as a measure of “intelligence”.

The server enabling in silico experiments, The Information Probability (IP) tool is available at https://ip.sano.science (accessed on 11 May 2023). This tool simulates the likelihood of a particular goal being achieved depending on the number of repetitions and elementary probability.

### 2.4. Higher Level of Organization

The degree of complexity with respect to a living organism as a system is reflected in one phenomenon: homeostasis. An organism can be characterized by two factors: 1. It is an open system and 2. Despite its open form, the organism ensures the stability of all components. The only solution that satisfies these conditions is a network of negative feedback loops that automatically stabilizes all system components without the need for external interference. The negative feedback loop involves the coupling of receptor and effector activity. The receptor is a specific structure that provides an answer to the question “How much?”. Meanwhile, the effector is a response to the question “How?” as in how to deliver the proper product. Exceeding the concentration conditions activates or deactivates the receptor. The signal sent by an active receptor to the effector activates the latter for as long as it continues to receive a signal from the receptor. The information transfer system, depending on the distance between these two centers, is either a concentration or an endocrine system with a precisely defined target. The tasks of the effector can also be performed by specialized cells that activate the processes connected with its specialized function as was mentioned above.

Simulating the layout of a system of related negative feedback loops would make it possible to track interdependencies. The deliberate and intentional introduction of disturbances would enable their effects to be tracked, which, in practice, could be a useful tool in drug design—especially with regard to the ‘management’ of this process—rather than in one that focuses on the defects of individual components [47,48,49].

All critical tasks on the negative feedback loop system pathway are carried out by proteins. The receptor holds the information about a specific interaction with a signaling molecule. It has the additional ability to change its structure through allosteric regulation. A receptor determines the course of further processes, e.g., the activation of an effector when such action is needed. It similarly deactivates the effector by not sending the signal molecule. All of these functions are fulfilled by proteins with a strictly defined activity characterized by high specificity. Receptors, which are most often proteins that are anchored in a membrane, attain their activity through appropriately shaping their structure based on the information of a sequence but also with the participation of the environment of the membrane itself. The degree of “intelligence”, i.e., the degree of a specific disorder in relation to the linear micelle-like arrangement, is very high in the case of receptor proteins. No negative feedback is fully independent. Signals sent from the organism modify the function of both the effector as well as the receptor. A receptor readjusting its sensitivity to a higher level responds to an external signal—due to reasons unknown to the given negative feedback unit. This is the strategy of communication of the organism in the form of sending a signal as a request. A detailed description with justification is provided in our publications [47,48,49].

The mechanisms described in this section can be tested using the following web applications that we have provided for the reader’s convenience: The NF Organized Systems (NFS—Negative Feedback System) tool is available at https://nfs.sano.science (accessed on 11 May 2023), with which the user can model interconnected systems and their interactions depending on system parameters, such as receptor sensitivity, effector reaction speed, or the time it takes for signals to travel between the receptor and the effector [47].

## 3. Discussion

In addition to determining the conditions of proteins that are folding, the external force field model is expressed for water influence (Equation (2)) and by other factors (Equation (7)) expressing the environment, which, when modified, provides various structures with a specific purpose. As was shown in a presentation of *Lactococcin 972* (PDB ID 2LGN [50]), when folded (using specialized programs) and according to *K* = 0, the protein appears to achieve a micelle-like structure with a specific hydrophobic core. The reality for this protein, however, is different. It clearly shows the influence exerted by a changed environment on the process of obtaining a protein with a specific function and how it results from the structure. The force fields used to predict the protein structure help to achieve a significantly improved structure for selected proteins. However, the same force field fails with another protein (examples in CASP [51]). The obvious conclusion is that such a highly diversified world of proteins (proteome) cannot be achieved by one common mechanism, i.e., a specific force field that takes into account only internal interactions. A significant role in the protein folding process is performed by the immediate environment that modifies the orientation of protein structuring as shown in the presented protein examples.

The water environment and the membrane environment are two examples of environmental differentiation: modification of the external force field. Other systems can also function as participants in terms of modifying the environment, including chaperones or chaperonins (manuscript in development).

The simulated functioning of a particular system, i.e., a living organism, which is based on the restoration of the homeostatic system, makes it possible to recreate this operation without the participation of any “boss” that is performing the management functions. It also facilitates adaptation to variable external factors. A higher level of organization, including in terms of communications between negative feedback loops, would not work as well, as if the individual goals were not achieved through p-based processes.

The details of the construction of the proteome, however, extend beyond the scope of the present publication and constitute a separate topic [47,48,49].

## 4. Conclusions

All processes taking place at the molecular level in a living organism can be interpreted in terms of energy flow. An analysis of information flow is no less important. The amount of information contained in 1 GB of a DNA record is remarkably small when compared with the final product that is created after processing the information contained in the DNA; further, it could be argued that the human body is the most complex system on our planet. The essence of the issue lies not in the amount of information available, but in how it is processed. Cell specialization is even associated with an additional reduction in information content by “silencing” certain genes, which thus results in cell specialization. At the translation stage, nature has a certain excess of information that is expressed in the degeneracy of the genetic code (whereby several triplets determine the same amino acid). On the other hand, the production of proteins that involves a synthesis of appropriate proteins with specific biological activity requires the contribution of external participants, including, in particular, those in the aquatic environment. This environment provides information supplementing the deficiency in the amino acid sequence → 3D structure stage, i.e., the biological function encoded in this structure. As a result of information processing at the proteins level, the proteome is already at an incomparably high level of complexity with respect to the source of information (DNA). The next few steps in the process involve the use of ready-made tools, whose arrangement and location play an important role. In terms of embryo development, this stage requires defining the directions of this process. This is achieved by determining the reference points in space, which is comparable to the actions of a blind tailor as has been discussed in [10,47].

Finally, the source of information and its processing as a programmed activity at every stage of biological activity at the molecular level needs to be determined. This processing provides the basis for the construction of a highly complex, living organism system that initially possesses a negligible amount of information (1 GB) in the DNA strand. An additional supplementary source of information is the environment, including, in particular, the water environment, which can be changed by different pH, with the presence of other molecules influencing the characteristics of water as it is. The latter is an important player providing significant variability in the protein’s structure design that complements the information deficiency at the protein folding stage. It concerns particularly the proteins, the structure of which is described by high *K* values. The 3D structure of proteins is a classic example of how a goal can be achieved on path “p” by defining the addressee through a specific notation (information) of micelle-like structural imperfections. During this stage, an already highly complex structure is achieved, one that ensures the entire system’s functioning [37,49,50,51]. The force field for internal interactions (inter-amino acids) is of fundamental importance. However, it is important to consider the different forms of the external field to understand and simulate the in silico folding process in a different environment. The information coded in the protein structure makes it possible to treat it as an “intelligent micelle”. It carries local discordance in a specific way versus the idealized micelle. Additionally, it is able to reach a specific goal primarily based on the p path [11,47,48,49,52].

The energy-dominated presentation of processes in living organisms requires complementation in the form of an information-based interpretation of biological phenomena. The highest deficiency of information in step 3 of biological dogma can be solved by treating the environment as the active participant in the protein folding process. The expression of the external force field, including water as well as other chemical compounds modifying the influence of water explains the differentiation of proteins’ status in respect to hydrophobicity distribution in the protein body. The proposed model of the external force field can also be used in protein folding simulations in silico.

The construction of the negative-feedback-loops network is suggested to simulate the higher order of organization (organism level) to express homeostasis characteristic for organisms.

## Figures and Tables

**Figure 1 entropy-25-00850-f001:**
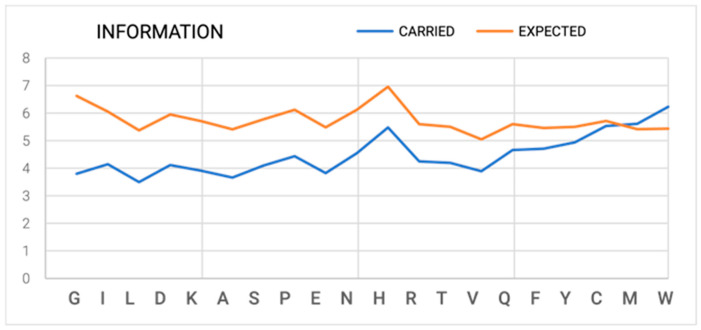
Quantity of information calculated according to Equation (1): Blue line—information carried by one amino acid, whereby the frequency of occurrence of a given amino acid in the non-redundant protein sub-base (“PDB-based”) [10] is taken into account; Orange line—the amount of information needed to identify a specific set of Phi and Psi angles (accuracy 5 deg × 5 deg) while taking into account the probability distribution (Ramachandran map—energy) for a given amino acid [10].

**Figure 2 entropy-25-00850-f002:**
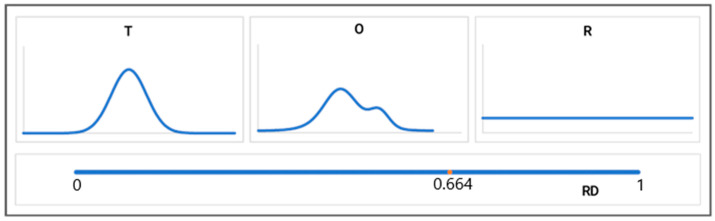
Visualization of the *T*, *O*, and *R* distributions together with the scale of Relative Distance (*RD*) measurements. *T* (**upper-left**) and *R* (**upper-right**) distributions in comparison with the *O* distribution (**upper-central**). Bottom—the *RD* scale with the position of the *O* distribution with an *RD* = 0.664 suggests a similarity to the *R* distribution rather than to the *T* distribution.

**Figure 3 entropy-25-00850-f003:**
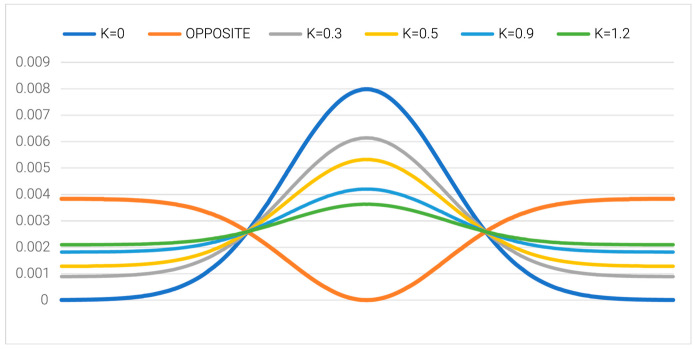
The representation of different forms of external force fields characterized by the value *K* as introduced in Equation (7). Dark blue line: Gaussian function and the external force field of pure water origin as well as the centric hydrophobic nucleus (i.e., the maximum hydrophobicity density in the center). Orange line: Opposite external force field with exposition of hydrophobicity on the surface and contact with the membrane’s hydrophobic environment. Other colors: The gradual modification of the *K* value (legend given on top).

**Figure 4 entropy-25-00850-f004:**
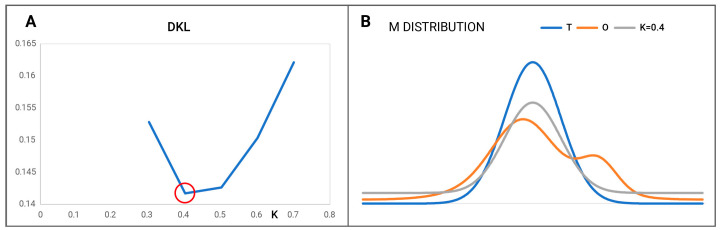
Visualization of the *M* distribution (according to Equation (7)). (**A**): The lowest *D_KL_* for (*O|M*) is obtained for *K* = 0.4. The best fit (the lowest *D_KL_* value) is obtained for *K* = 0.2 distinguished by red circle. This value of *K* generates the closest *M* distribution versus the *O* distribution. This is interpreted as the best to represent the modified *T* distribution for the *O* distribution. (**B**): The distributions are shown in Figure 2 with the *M* distribution present (grey).

**Figure 5 entropy-25-00850-f005:**
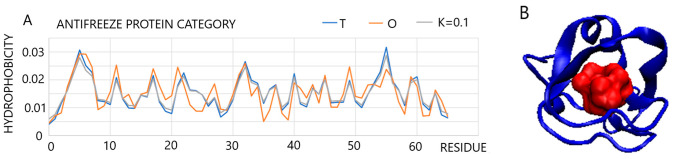
Characteristics of antifreeze protein with low *K* value, i.e., 0.1. (**A**): Set of *T*, *O*, and *M* profiles for a protein representing a micelle-like structure. (**B**): 3D presentation of the structure with red residues distinguished representing hydrophobic core built by the residues of both high (above 0.02) *T_i_* and *O_i_* values on the profiles.

**Figure 6 entropy-25-00850-f006:**
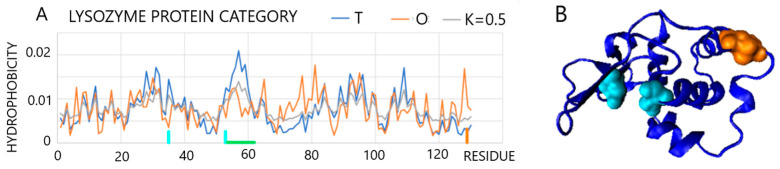
Characteristics of lysozyme: (**A**): Profiles representing *T* (red), *O* (blue), and *M* (gray) distributions for *K* = 0.5 with local discrepancy distinguished for fragment indicated by cyan horizontal line. Positions of catalytic residues are represented by cyan vertical lines, and the position of 128Cys is distinguished on *x*-axis. (**B**)—3D presentation with residues distinguished as shown in (**A**).

**Figure 7 entropy-25-00850-f007:**
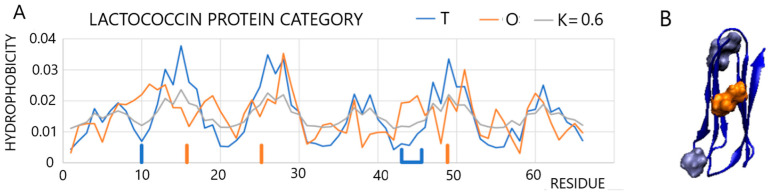
Characteristics of protein active in the periplasm. (**A**): profiles *T*, *O*, and *M* for *K* = 0.6. Highlighted residues: Orange—expected hydrophobic core with high *T_i_* and *O_i_* values, where the *Oi* values are much lower; the residues distinguished by blue vertical and horizontal lines represent significant discrepancy between *O* and *T* distributions. (**B**): 3D presentation with orange residues representing deficiency of hydrophobicity and blue ones representing excess of hydrophobicity. The distinguished residues as shown in **A**.

**Figure 8 entropy-25-00850-f008:**
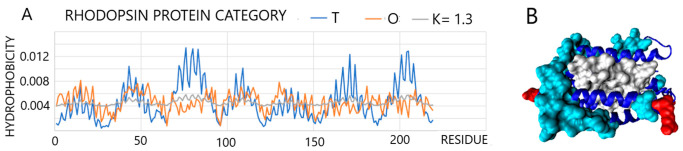
Characteristics of transmembrane protein rhodopsin: (**A**): Profiles *T*, *O*, and *M* for *K* = 1.3. (**B**): 3D presentation with highlighted residues: Red: Residues with *Ti* and *Oi* hydrophobicity; cyan residues represent the excess of hydrophobicity on the protein surface, while white residues are those that represent the expected hydrophobic nucleus (*Ti* high) that is not the case (low *Oi*).

**Figure 9 entropy-25-00850-f009:**
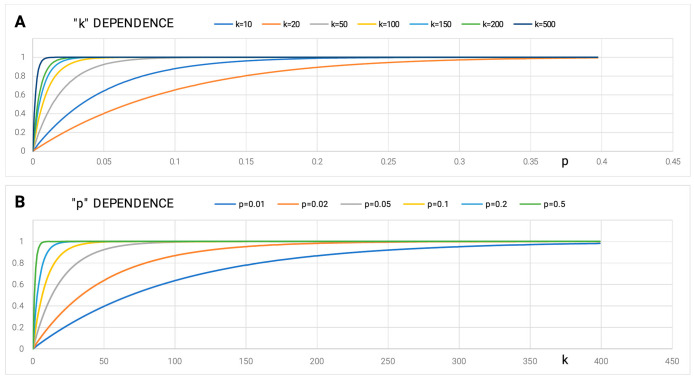
Reaching a goal using the discussed model via *p* values depending on p and k. (**A**)—Dependence on p with an increase in the value of k; (**B**)—Dependence on k with an increase in the value of *p*.

## Data Availability

Not applicable.

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
