# Peer review of "Protein Is an Intelligent Micelle"

_entropy, 2023, doi:10.3390/e25060850_

Round 1

Reviewer 1 Report (New Reviewer)

In the article “Protein is an intelligent micelle”, the authors discuss the amount of information requested for a protein to perform the task it has been evolutionary determined to do.

I was asked to review this article after a first round of review/corrections has been done. From red parts in the text, I can guess that important corrections/additions have been performed. However, they do not prevent concerns about the ms. 

The topic is interesting but the type of the present manuscript is not well defined. It is not a “true” research article or review. Here, the authors present mainly comments/reflections on the information flow in biological processes as a research article. The resulting manuscript is puzzling and lacks consistency.

Below are a few major comments:

-          The introduction does not give clues on the following text; it gives only generalities on information.

-          The FOD and FOD-M models are abruptly introduced in the Materials and Methods (p 2) with missing definitions.   

-          The information on the models is difficult to understand. I found clear and full details in an article from the same authors (Roterman et al, PlosOne 2022). Here there is an attempt to present them more briefly, but the result lacks clarity. 

-          In the present text, it is unclear whether deviation from FOD model provides information on folding or on performing a task after folding. The examples given by the authors in Fig. 4 to 7 provide information on the tasks performed by the protein. In this case, how does the environment provide information for proper folding? This step is missing.

-          Most “Results” are either general comments on information (3.1 to 3.4, 3.7) or comments on functional information contained in 3D structures from hydrophobicity indexes (3.5).

 Minor comments: The text is full of editorial mistakes that should be corrected.

In summary, the manuscript might offer an interesting reflection on information in the field of structural biology. Nevertheless, it is not fitted for publication in its present form. I would recommend to strongly reduce its length and to rewrite it as an essay or a perspective article, limited to 3000 words.

Author Response

REVIEWER I

Open Review

English language and style

( ) English is very difficult to understand/incomprehensible
( ) Extensive editing of English language and style required
( ) Moderate English changes required
(x) English language and style are fine/minor spell check required
( ) I don't feel qualified to judge about the English language and style

Yes

Can be improved

Must be improved

Not applicable

Does the introduction provide sufficient background and include all relevant references?

( )

( )

(x)

( )

Are all the cited references relevant to the research?

(x)

( )

( )

( )

Is the research design appropriate?

( )

( )

(x)

( )

Are the methods adequately described?

( )

( )

(x)

( )

Are the results clearly presented?

( )

( )

(x)

( )

Are the conclusions supported by the results?

( )

( )

(x)

( )

Comments and Suggestions for Authors

In the article “Protein is an intelligent micelle”, the authors discuss the amount of information requested for a protein to perform the task it has been evolutionarily determined to do.

I was asked to review this article after a first round of review/corrections has been done. From the red parts in the text, I can guess that important corrections/additions have been performed. However, they do not prevent concerns about the ms. 

The topic is interesting but the type of the present manuscript is not well-defined. It is not a “true” research article or review. Here, the authors mainly present comments/reflections on the information flow in biological processes as a research article. The resulting manuscript is puzzling and lacks consistency.

Below are a few major comments:

-          The introduction does not give clues on the following text; it gives only generalities on the information.

The INTRODUCTION was changed.

-          The FOD and FOD-M models are abruptly introduced in the Materials and Methods (p 2) with missing definitions.

The presentation was extended.

-          The information on the models is difficult to understand. I found clear and full details in an article from the same authors (Roterman et al, PlosOne 2022). Here there is an attempt to present them more briefly, but the result lacks clarity.

The shortening of the model presentation is to avoid redundancy in the presentation of the FOD and FOD-M models.

 -          In the present text, it is unclear whether deviation from the FOD model provides information on folding or on performing a task after folding. The examples given by the authors in Fig. 4 to 7 provide information on the tasks performed by the protein. In this case, how does the environment provide information for proper folding? This step is missing.

The explanation was added.

The proteins of structures according to the 3D Gauss distribution of hydrophobicity represent the effect of the water environment – the external force field in water conditions. The M profile obtained after introducing the presence of opposite characteristics to polar water represents the effect of an external force field modified by K. The membrane protein structure is accordant to high “presence” – modification of the polar surrounding by a hydrophobic component. The M profile is assumed to represent the specificity of the environment. The value of K measures the degree of this modification. The O profile is generated according to environmental influence on the folding chain. The M distribution is the T distribution for other than water surrounding.

-          Most “Results” are either general comments on information (3.1 to 3.4, 3.7) or comments on functional information contained in 3D structures from hydrophobicity indexes (3.5).

The ideal micelle construction is able to do one single action – high solubility and occasional random interaction with ions or polar compounds. The specificity (read information) is coded in the specific local discordance of micelle-like distribution. For example – the local deficiency of hydrophobicity is mostly caused by the presence of a cavity. The identification of residues representing hydrophobicity deficiency can be identified as localized in the cavity. This is shown in lysozyme and other enzymes (especially one-chain enzymes). The local excess of hydrophobicity – especially on the surface, suggests the place ready for protein-protein complexation (see Dygut et all Structural Interface Forms and Their Involvement in Stabilization of Multidomain Proteins or Protein Complexes. Int J Mol Sci. 2016;17(10):1741. doi: 10.3390/ijms17101741).

The presented examples of proteins visualise the influence of the environment (increased K values) on the structure which is constructed according to a modified (by K) force field – different from the water environment.

The aim of this presentation: the In Silico folding simulation shall take into account the external conditions (water modified by other factors). Tracing the results of the CASP experiment – the very good program (force field) delivers a structure highly accordant with the target, while the same very good force field (same program) delivers bad results. The conclusion is – the protein was folded in other conditions compared to the first one. The force field (program) is OK, but there is a difference in the protein folding mechanism.

Minor comments: The text is full of editorial mistakes that should be corrected.

The current version of the manuscript is after language correction.

In summary, the manuscript might offer an interesting reflection on information in the field of structural biology. Nevertheless, it is not fitted for publication in its present form. I would recommend to strongly reduce its length and to rewrite it as an essay or a perspective article, limited to 3000 words.

The status of the paper has been changed from “article” to “review”.

It is difficult to make the paper shorter satisfying the above comment about the need for a more detailed model description.

Reviewer 2 Report (Previous Reviewer 1)

This is a revised version of the original document. A few things have been fixed, but a number of issues remain and the paper will be hard for readers to make use of.

Significant:

1) The title is still totally mismatched with the paper. There is no discussion of what definition of intelligence is being used here, or how the data speak to the intelligence of a protein. Intelligence is only mentioned once in the body of the paper. It is a provocative title, which is fine, but then the content has to support the claim. I don't see the current content addressing at all the question of intelligence. To be clear: I don't dispute that one *could* give an account of protein intelligence - I'm not against this idea - but this paper doesn't cover that at all. The authors had replied "The definition promoted here is: a protein knows its purpose" but again, this needs a lot of unpacking - the definition of "knows" and "purpose" is not easy to give in the context of a protein. It requires specific work for readers to take this seriously.  I think the title will be seen as completely misleading by readers if the authors don't address this more directly.

More minor:

2) The English is still not optimal - it uses some awkward constructions in places. This is not a fatal issue, but it can certainly be improved.

Author Response

REVIEWER II

Open Review

English language and style

( ) English very difficult to understand/incomprehensible
( ) Extensive editing of English language and style required
(x) Moderate English changes required
( ) English language and style are fine/minor spell check required
( ) I don't feel qualified to judge about the English language and style

Yes

Can be improved

Must be improved

Not applicable

Does the introduction provide sufficient background and include all relevant references?

( )

(x)

( )

( )

Are all the cited references relevant to the research?

(x)

( )

( )

( )

Is the research design appropriate?

(x)

( )

( )

( )

Are the methods adequately described?

( )

(x)

( )

( )

Are the results clearly presented?

( )

(x)

( )

( )

Are the conclusions supported by the results?

( )

(x)

( )

( )

Comments and Suggestions for Authors

This is a revised version of the original document. A few things have been fixed, but a number of issues remain and the paper will be hard for readers to make use of.

Significant:

  • The title is still totally mismatched with the paper. There is no discussion of what definition of intelligence is being used here, or how the data speak to the intelligence of a protein. Intelligence is only mentioned once in the body of the paper. It is a provocative title, which is fine, but then the content has to support the claim. I don't see the current content addressing at all the question of intelligence. To be clear: I don't dispute that one *could* give an account of protein intelligence - I'm not against this idea - but this paper doesn't cover that at all. The authors had replied "The definition promoted here is: a protein knows its purpose" but again, this needs a lot of unpacking - the definition of "knows" and "purpose" is not easy to give in the context of a protein. It requires specific work for readers to take this seriously.  I think the title will be seen as completely misleading by readers if the authors don't address this more directly.

The ideal micelle construction (central hydrophobic core with a polar surface) is able to do one single action – high solubility with occasional random interaction with ions or polar compounds. The specificity (read “information”) is coded in the specific local discordance of micelle-like distribution. For example – the local deficiency of hydrophobicity is mostly caused by the presence of a cavity. The identification of residues representing hydrophobicity deficiency can be identified as localized in the cavity. It is shown in lysozyme and other enzymes (especially one-chain enzymes). The local excess of hydrophobicity – especially on the surface suggests the place ready for protein-protein complexation (see Dygut et all Structural Interface Forms and Their Involvement in Stabilization of Multidomain Proteins or Protein Complexes. Int J Mol Sci. 2016;17(10):1741. doi: 10.3390/ijms17101741).

The presented examples of proteins visualise the influence of the environment (increased K values) on the structure which is constructed according to a modified (by K) force field – different from the water environment.

The aim of this presentation: the In Silico folding simulation shall take into account the external conditions (water modified by other factors). Tracing the results of CASP experiment – the very good program (force field) delivers a structure highly accordant with the target, and the same very good force field (same program) delivers bad results. The conclusion is – the protein was folded in other conditions compared to the first one. The force field (program) is OK, but the protein folding mechanism differs.

Generally speaking – the „specificity” (often used in biochemistry manuals) shall be substituted by the expression “information-carrying”. The specificity is treated as the intelligence of protein. 

More minor:

2) The English is still not optimal - it uses some awkward constructions in places. This is not a fatal issue, but it can certainly be improved.

The text of the current version of the ms has been corrected by specialists – the office with the licence.

Reviewer 3 Report (New Reviewer)

The manuscript of Roterman et al. “Protein is an intelligent micelle” is aimed to describe how the specific information encoded in the living organisms is proceeded at different molecular levels. The authors focus on the concept of the protein representation as a “intelligent micelle” which is organized in a way that the functional protein residues stand out from the hydrophobic/hydrophilic profile of the “ideal micelle”. In particular, these residues that do not follow the rule of ideal hydrophobicity distribution within the protein globule carry the most significant specific information about the protein function and its role in the ongoing biologically relevant processes in the cell. The manuscript consists of many curious examples and analogies that help to get original philosophical insights into the basics of life and its molecular organization. Although the topic of the paper is relevant for a publication in Entropy, the way the manuscript is organized is sometimes obscure, it requires significant work on the formulations, the use of broadly accepted concepts and the language quality. One of the central concerns is that this manuscript was submitted as “Article” instead of “Review”. The concepts discussed here have been developed by the authors for years, and the manuscript, at least in its actual form, does not include any novel data but rather summarizes the knowledge obtained by the authors in the years of development of their concepts. Therefore, it would be strongly suggested that the manuscript is transferred into the “Review” type of publication when newly submitted.

MAJOR POINTS

POINT 1. The authors use the concept of “force field” erroneously through the whole manuscript.

POINT 2. It would be useful to prove the data on the probabilities of the nucleotides in the DNA (e.g. for human genome) as well as for the amino acid residues (again, e.g. for human) in all proteins.

POINT 3. Why do authors consider negative feedback loops as the only solution in the context they mean it? It is not appropriately justified.

POINT 4. The whole subsection 3.7 is written in a very unclear and obscure way, both in terms of the content and the language. It should be completely rewritten.

POINT 5. The same applies for the first paragraph of the discussion section.

POINT 6. Line 601: “The human body is the most complex system on our planet” is very much arguable.

MINOR POINTS

In general, the manuscript suffers immensely from the the grammatical mistakes and typos, which makes it tiresome and irritating for a potential reader, and, therefore, the English usage should be essentially edited to improve the quality of the text before it could be considered for acceptance in the journal. Apart from this, there are parts of the manuscript starting with “Ad. 1”, “Ad. 2” etc., and multiple parts of the text are in red. Could it be that the authors did not upload the most recent version of their manuscript but a working draft instead? The use of “-” and “– ” through the manuscript if very often improper.

– Line 34: “syem”.

– “Materials and Methods”: in the list, there are sometimes full points at the end of the items, and sometimes there are no.

– Line 81: the K parameter needs to be introduced properly.

– Line 87: “a non-redundant PDB” => “a non-redundant dataset from the PDB”.

– Line 93: “protein base” => “protein structure database”.

– Line 97: “Phi I Psi” => “Phi and Psi”.

– Line 108: inconsistent format.

– Line 113: “factor that actively participates” is an unfortunate formulation.

– Line 116: “is not recognized outside the structure of ice” is an unfortunate formulation.

– Line 120 and other similar usage mistakes: “The participation of an external force field”.

– Line 135: “).” = “)”.

– Line 152: “relation” => “relation,”.

– Line 172: Starting a sentence with “And” is not appropriate.

– Line 196: “degeree” => “degree”.

– Line 225: Figure 1 caption: “of a field generated by water” is an unfortunate formulation.

– Line 250: “It is here that the problem” is an unfortunate formulation.

– Line 268: “maschines” => “machines”.

– Line 268: “Paricular” => “Particular”.

– Line 270: “respeonsible” => “repsonsible”.

– Line 272: “proteisn” => “proteins”

– Line 289: “Man, despite his ability to solve many problems- is unable” is an unfortunate formulation.

– Line 295: the sentence is unclear.

– Line 298: “Planet” is used, later “planet” is used.

– Lines 332-333: is an unfortunate formulation.

– Line 343: “sent” => “sold”.

– Line 350: “unethical,)” => “unethical)”.

– Figure 2 caption: both “k” and “K” are used in the same context.

– Lines 393-395: the sentence is unclear.

– Line 409: “sufficient” for which purpose?

– Line 413: “guarantee” => “guarantees”.

– Lines 453-455: the sentence is unclear.

– Figure 5 caption: “elimitatio” => “elimination”.

– Figure 6 caption: “residuals” => “residues”.

– Lines 537-538: “structure is a basic factor” is an unfortunate formulation.

– Lines 568-569: “the external force field model expressed by means of Mi” is an unfortunate formulation.

– Line 609: a redundant space should be removed.

– Line 610: a redundant space should be removed.

– Line 620: “construction” => “construction of”.

– Line 633: “cocded” => probably the authors mean “encoded”.

– Line 634: “posssible” = “possible”.

Author Response

REVIEWER III

Open Review

English language and style

(x) English very difficult to understand/incomprehensible
( ) Extensive editing of English language and style required
( ) Moderate English changes required
( ) English language and style are fine/minor spell check required
( ) I don't feel qualified to judge about the English language and style

Yes

Can be improved

Must be improved

Not applicable

Does the introduction provide sufficient background and include all relevant references?

( )

( )

(x)

( )

Are all the cited references relevant to the research?

(x)

( )

( )

( )

Is the research design appropriate?

( )

( )

(x)

( )

Are the methods adequately described?

( )

( )

(x)

( )

Are the results clearly presented?

( )

( )

(x)

( )

Are the conclusions supported by the results?

( )

( )

(x)

( )

Comments and Suggestions for Authors

The manuscript of Roterman et al. “Protein is an intelligent micelle” is aimed to describe how the specific information encoded in the living organisms is proceeded at different molecular levels. The authors focus on the concept of the protein representation as a “intelligent micelle” which is organized in a way that the functional protein residues stand out from the hydrophobic/hydrophilic profile of the “ideal micelle”. In particular, these residues that do not follow the rule of ideal hydrophobicity distribution within the protein globule carry the most significant specific information about the protein function and its role in the ongoing biologically relevant processes in the cell. The manuscript consists of many curious examples and analogies that help to get original philosophical insights into the basics of life and its molecular organization. Although the topic of the paper is relevant for a publication in Entropy, the way the manuscript is organized is sometimes obscure, it requires significant work on the formulations, the use of broadly accepted concepts and the language quality. One of the central concerns is that this manuscript was submitted as “Article” instead of “Review”. The concepts discussed here have been developed by the authors for years, and the manuscript, at least in its actual form, does not include any novel data but rather summarizes the knowledge obtained by the authors in the years of development of their concepts. Therefore, it would be strongly suggested that the manuscript is transferred into the “Review” type of publication when newly submitted.

The current status of the ms is “Review” instead of “Article”.

MAJOR POINTS

POINT 1. The authors use the concept of “force field” erroneously throughout the whole manuscript.

The term “force field” is understood as a set of influences/forces leading to a specific arrangement within the protein structures. This term contains components which, according to the Authors of, e.g. the given package consider relevant: electrostatic, vdW, H-bonds, and those which take into account the possibility of a flexible structure (stretching, bending and other vibrations). Force fields applied in different packages differ in the record of the force field concerning internal influences within the polypeptide chain. Force fields take into account the presence of water by introducing the appropriate number of water particles (depending on the size of the box in which the folded protein is placed). Water interacts according to an atom-atom interaction principle. In the FOD mode, a water-based environment (or another modified environment) is treated as a continuum – in the form of an external force field, whose presence is expressed by directing the folding process towards a structure of a varying degree of reproduction of the micelle-like form.

POINT 2. It would be useful to prove the data on the probabilities of the nucleotides in the DNA (e.g. for human genome) as well as for the amino acid residues (again, e.g. for human) in all proteins.

A detailed analysis within the scope of the complete genome can provide information on the frequency of occurrence of the given nucleotide, e.g. depending on the organism – especially when it comes to a comparison from an evolutionary perspective. Programs identifying the location of genes are based on the principle of nucleotide occurrence frequency different than random. Because in the human genome, the gene record constitutes approximately 3% of the entire DNA strand, it is not a huge leap to assume a random structure for the entire genome.

An analysis proposed in this section limited to the human organism would most likely be a very interesting study. Such research is conducted by genomics specialists. For the purpose of this article, the presented level of detail seems to be appropriate.

POINT 3. Why do authors consider negative feedback loops as the only solution in the context they mean it? It is not appropriately justified.

The primary premise for such an implementation is the phenomenon defined as homeostasis. It involves maintaining a state of balance in a significant majority of processes.

It is possible here to use a very simple example.

Below is a fragment of a set of test results of blood drawn at any given medical analysis laboratory.

The most significant part of this compilation is the right-side column. It signifies that the concentration levels of individual (all) components should be within the ranges specified in that column. What the existence of such ranges by itself says about the structure is that it can be a question of chance. Thus, there is a system that sees to it that the ranges of a HEALTHY human are maintained. The items in the presented compilation highlighted with the letter H and L signify states of exceeding the correct ranges. This means no more and no less than the fact that the negative feedback cycle responsible for stabilising the production of a given component is not functioning properly. Exceeding the proper ranges leads to the question concerning its causes: a receptor has failed because it did not react properly to heightened levels (failed to send information to deactivate the effector), or the effector did not receive the signalling molecule informing of its deactivation. It could also mean that the effector itself is not functioning correctly by not reacting properly to the information transmitted to it.

Of course, adopting a set of negative feedback cycles is intended to simulate the correct condition. The pathology phenomenon in the In Silico experiment should involve the targeted introduction of a disruption in the form of a change in the parameters of the receptor or effector, or the system of communication between negative feedback units, tracing the effects of the purposefully introduced disruptions. Such an In Silico experiment is, of course, possible only for a properly constructed pathogen.

The issue is discussed in detail in the publication: Konieczny L, Roterman I, Spolnik P – Systems Biology – Springer 2014.

POINT 4. The whole subsection 3.7 is written in a very unclear and obscure way, both in terms of the content and the language. It should be completely rewritten.

The subsection has been supplemented. Its clarification is available in the literature stated above. An electronic version of this handbook is currently in development and should become available online within several months.

POINT 5. The same applies to the first paragraph of the discussion section.

The subsection has been corrected. Additionally, the additions made to the main body of the article should clarify any doubts contained in this subsection.

POINT 6. Line 601: “The human body is the most complex system on our planet” is very much arguable.

“The human body seems to be the most complex system on our planet” – would this form be more agreeable to you? The questions – do you know anything else more complicated and effectively working correctly about 60 years?

This, of course, is an opinion. In the article, the status of which has been changed to a “review”, I believe such a statement would be admissible.

MINOR POINTS

In general, the manuscript suffers immensely from the the grammatical mistakes and typos, which makes it tiresome and irritating for a potential reader, and, therefore, the English usage should be essentially edited to improve the quality of the text before it could be considered for acceptance in the journal. Apart from this, there are parts of the manuscript starting with “Ad. 1”, “Ad. 2” etc., and multiple parts of the text are in red. Could it be that the authors did not upload the most recent version of their manuscript but a working draft instead? The use of “-” and “– ” through the manuscript if very often improper.

The entire text has been edited. Additionally, a different translation service agency has been employed. Thank you kindly for the thorough analysis. We truly appreciate the effort undertaken by the Reviewer.

– Line 34: “syem”.

– “Materials and Methods”: in the list, there are sometimes full points at the end of the items, and sometimes there are no.

– Line 81: the K parameter needs to be introduced properly.

– Line 87: “a non-redundant PDB” => “a non-redundant dataset from the PDB”.

– Line 93: “protein base” => “protein structure database”.

– Line 97: “Phi I Psi” => “Phi and Psi”.

– Line 108: inconsistent format.

– Line 113: “factor that actively participates” is an unfortunate formulation.

– Line 116: “is not recognized outside the structure of ice” is an unfortunate formulation.

– Line 120 and other similar usage mistakes: “The participation of an external force field”.

– Line 135: “).” = “)”.

– Line 152: “relation” => “relation,”.

– Line 172: Starting a sentence with “And” is not appropriate.

– Line 196: “degeree” => “degree”.

– Line 225: Figure 1 caption: “of a field generated by water” is an unfortunate formulation.

– Line 250: “It is here that the problem” is an unfortunate formulation.

– Line 268: “maschines” => “machines”.

– Line 268: “Paricular” => “Particular”.

– Line 270: “respeonsible” => “repsonsible”.

– Line 272: “proteisn” => “proteins”

– Line 289: “Man, despite his ability to solve many problems- is unable” is an unfortunate formulation.

– Line 295: the sentence is unclear.

– Line 298: “Planet” is used, later “planet” is used.

– Lines 332-333: is an unfortunate formulation.

– Line 343: “sent” => “sold”.

– Line 350: “unethical,)” => “unethical)”.

– Figure 2 caption: both “k” and “K” are used in the same context.

– Lines 393-395: the sentence is unclear.

– Line 409: “sufficient” for which purpose?

– Line 413: “guarantee” => “guarantees”.

– Lines 453-455: the sentence is unclear.

– Figure 5 caption: “elimitatio” => “elimination”.

– Figure 6 caption: “residuals” => “residues”.

– Lines 537-538: “structure is a basic factor” is an unfortunate formulation.

– Lines 568-569: “the external force field model expressed by means of Mi” is an unfortunate formulation.

– Line 609: a redundant space should be removed.

– Line 610: a redundant space should be removed.

– Line 620: “construction” => “construction of”.

– Line 633: “cocded” => probably the authors mean “encoded”.

– Line 634: “posssible” = “possible”.

Round 2

Reviewer 1 Report (New Reviewer)

The authors fulfilled my concerns about the manuscript that has been much improved. The revised version is easy to understand. The authors'approach is clearly explained and of broad interest for readers.

Author Response

Many thanks for valuable comments. I really appreciate the big effort to make our paper upgraded.

Many thanks once again.

Reviewer 2 Report (Previous Reviewer 1)

I thank the authors for engaging with the comments and greatly improving the manuscript. It's a very interesting approach and I am happy to see it published.

Author Response

Many thanks for valuable comments. I really appreciate the big effort to make our paper upgraded.

Many thanks once again.

Reviewer 3 Report (New Reviewer)

The authors addressed all the concerns from the first review.

Author Response

Many thanks for valuable comments. I really appreciate the big effort to make our paper upgraded.

Many thanks once again.

This manuscript is a resubmission of an earlier submission. The following is a list of the peer review reports and author responses from that submission.

Round 1

Reviewer 1 Report

- the Introduction needs a lot of improvement. It has some details but is missing the big picture for the reader - what problem is going to be addressed?  Why is it important?  Before jumping into formulas, explain what question is going to be discussed and how it's being approached (and with what goal). Also, information in biology is certainly not limited to DNA.

- the title doesn't match the Abstract at all.

- This paper attempts to characterize the "intelligence" behind the specificity of proteins, and how the micro environment may be requisite for its germination, using basic information-theoretic measures. The paper does a good job of introducing the basic ideas, namely the "k" and "p" strategies to solve problems. The rest of the paper essentially describes how proteins may have come to adopt a relatively more intelligent strategy, namely "p", to solve the problem of substrate-binding specificity in biological systems. The main argument is that the information deficit between the amino acid sequence and the conformational shape of the protein post translation is bridged by the information supplied by the external micro environment (water) that generates a "force field" guiding the folding of the protein into a micelle-like overall structure (polar surface with a hydrophobic nucleus, for example). While this is a resonable argument, it does not support the overall conclusion that the *main* player in the determination of the protein structure is the external environment rather than the internal interactions. Such a conclusion would require more concrete quantifications of the ratio of the external-to-internal information that closes the sequence-to-shape information deficit. Hence, I feel that the authors should either consider including a quantification of the role of internal interactions relative to that of the external environment in support of its present conclusions, or modify the conclusions to simply acknowledge the role of the environment in protein folding rather than tout it as the main player.    A few miscellaneous comments: - Might it possible to quantify the amount of information contributed by the environment (water) to the protein structure? To what extent would it close the gap shown in Fig 2? - The use of the term "intelligence" may be misplaced, where it's equated to the development of specificity, as there's no discussion of the possibility of adaptation to novel circumstances. There has been a lot of discussion of intelligence with many possible definitions besides this one. Perhaps it could be better understood as evolutionary intelligence, not lifetime intelligence? Another difference compared to the traditional usage of the term is that there seems to be no feedback from the protein channeled through the environment back to itself, which is often thought of as an essential component of the cognition of "agents" in the cognitive science literature. Perhaps a protein does not exhibit agent-like intelligence in this sense, but is more of a proto-intelligent entity that's equipped with the necessary components of basal cognition? - The section (3.7) on homeostasis seems to be rather disconnected from the flow of the rest of the paper. Specifically, it's not clear how homeostasis is connected to the specificity of the protein. It in fact comes across as something opposite (perhaps complementary) to the local-disorder-character of the protein that gives its specificity.

- the paper requires a spell check/grammar editing - for example, 1st word of Abstract - "Interpretating" - should be "Interpreting". There are many examples of English constructions that should be improved. Line 44, "used" should be "user". Line 439, should be "proteome" not "proteom". Many more like this. The writing can really be improved. Also, the language style used in the description of the software tool (https://hphob.sano.science/) may be adopted in the main paper, as it's more clear and succinct.

Reviewer 2 Report

The article “Protein is the intelligent micelle” by Roterman and Konieczny describes the “protein” significant relations and roles in the biological system. They use different online sources to explain their theory. This manuscript is more or less written like an essay or story. They gave a lot of unnecessary examples which out of context. Authors should thoroughly revise the manuscript and carefully write according to the aim. Illogical or nonscientific samples should be removed. Some of the information provided is very basic; no need to write definitions. Please write everything to the point.  

The quality of the figures is poor, and authors should use the same font and text format. Also, all figures should be labeled appropriately. 

The discussion section is weak and should be rewritten, especially discussing the findings of their analysis and comparing them with previously published data. 

The limitations of the should also be highlighted. 

There are a lot of grammatical and other language errors; please correct them.